# Anticipating Leucovorin Rescue Therapy in Patients with Osteosarcoma through Methotrexate Population Pharmacokinetic Model

**DOI:** 10.3390/pharmaceutics16091180

**Published:** 2024-09-06

**Authors:** Laura Ben Olivo, Pricilla de Oliveira Henz, Sophia Wermann, Bruna Bernar Dias, Gabriel Osorio Porto, Amanda Valle Pinhatti, Manoela Domingues Martins, Lauro José Gregianin, Teresa Dalla Costa, Bibiana Verlindo de Araújo

**Affiliations:** 1Pharmacokinetics and PK/PD Modeling Laboratory, Pharmaceutical Sciences Graduate Program, Federal University of Rio Grande do Sul, 2752 Ipiranga Ave., Santana, Porto Alegre 90610-000, RS, Brazil; laura.olivo@ufrgs.br (L.B.O.); pricilla.henz@gmail.com (P.d.O.H.); sophiawermann@gmail.com (S.W.); b.bernardias@gmail.com (B.B.D.); gabriel.osorioporto@gmail.com (G.O.P.); dalla.costa@ufrgs.br (T.D.C.); 2Medical Sciences Graduate Program, Federal University of Rio Grande do Sul, Porto Alegre 90610-000, RS, Brazil; avpinhatti@hcpa.edu.br; 3Faculty of Dentistry, Federal University of Rio Grande do Sul, Porto Alegre 90610-000, RS, Brazil; manomartins@gmail.com; 4Pediatric Oncology Service, Hospital de Clínicas de Porto Alegre, Department of Pediatrics, Federal University of Rio Grande do Sul, Porto Alegre 90610-000, RS, Brazil; lgregianin@gmail.com

**Keywords:** methotrexate, osteosarcoma, pharmacokinetic model, Brazilian pediatric patients

## Abstract

Methotrexate (MTX), which presents high inter-individual variability, is part of the Brazilian Osteosarcoma Treatment Group (BOTG) protocol. This work aimed to develop a MTX population pharmacokinetic model (POPPK) for Brazilian children with osteosarcoma (OS) following the BOTG protocol to guide rescue therapy and avoid toxicity. The model was developed in NONMEM 7.4 (Icon^®^) using retrospective sparse data from MTX therapeutic drug monitoring of children attending a southern Brazilian public reference hospital. Data were described by a two-compartment model using 216 MTX cycles from 32 patients (5–18 y.o.) with OS who received 12 g/m^2^ dose/cycle. To explain inter-individual and inter-occasion variability in clearance and peripheral volume, covariates from demographic and biochemical data were evaluated. Serum creatinine was a significant covariate of MTX clearance (14.8 L/h), and the body surface area (BSA) was significant for central compartment volume (82.5 L). Inter-compartmental clearance and volume of peripheral compartment were 0.178 L/h and 5.72 L, respectively. The model adequately describes MTX exposure in Brazilian children with OS. Successful simulations were performed to predict MTX concentrations in pediatric patients above five years old with acute kidney injury and anticipate rescue therapy adjustments.

## 1. Introduction

Osteosarcoma (OS), the prevalent primary malignant bone tumor, is found globally at an approximate rate of one to three cases per million individuals annually [1]. The standard treatment for OS consists of neoadjuvant chemotherapy before surgery, surgical resection, and adjuvant chemotherapy in the postoperative period [2]. This schedule of treatment was first introduced by Rosen and coworkers in 1979. Back then, the protocol consisted of the administration of high-dose methotrexate (HDMTX) in combination with adriamycin and cyclophosphamide [3].

Hospitals in countries such as France and Germany, and some American hospitals, have incorporated etoposide and isosfamide in their protocol to reduce the use of HDMTX [4]. However, in Brazil, HDMTX is still the gold standard for OS treatments, and it is combined with cisplatin (CIS) and doxorubicin (DOX), in so-called MAP therapy. Despite some trials with non-HDMTX conducted in Brazil, the local public health system (Sistema Único de Saúde—SUS) still considers MAP therapy as OS’s first line of treatment [5]. The Brazilian Osteosarcoma Treatment Group (BOTG) has established a protocol for OS which involves administering MAP chemotherapy for 1 to 10 weeks, followed by surgery in week 11 or 12, and repeating the 10-week cycle after surgery [5,6].

The administration of HDMTX is associated with toxicity, and therefore, therapeutic monitoring of serum methotrexate (MTX) concentrations is necessary during its use [7]. Several adverse effects are associated with MTX toxicity levels. The most common adverse effects are neutropenia, any grade of mucositis, thrombocytopenia, and neurotoxicity. The occurrence of toxicity is diminished due to the traditional therapeutic drug monitoring (TDM) of MTX serum concentrations and the application of suitable folinic acid (leucovorin, LCV) rescue, coupled with proper patient hydration and urine alkalization [8]. Despite MTX monitoring, rescue therapy adjustments in Brazilian institutions rely on empirical methods.

Monitoring MTX concentrations can greatly aid patient management, given that the effectiveness of HDMTX has been demonstrated to rely on a C_max_ level of 700 to 1000 μM after a 4- to 6-h infusion [9,10]. On the other hand, it is recommended for cytotoxic drugs that the monitoring of serum levels should consist of the evaluation of the extension of drug exposure, which is given by the area under the serum concentration–time curve (AUC) [11]. Therefore, the safe range for the MTX AUC was found to be between 4000 and 12,000 μM.h [10,12].

Despite the determination of this safe range, pediatric patients still present high inter-individual variability in MTX levels. Model-informed precision dosing (MIPD) is used when algorithms are integrated to predict the best dosage regimen for a given patient, improving the chance of a better outcome than the standard protocol. These algorithms are usually based on population analysis and consider patients’ specific characteristics [13]. Many studies have reported MTX pharmacokinetic (PK) in children with OS. In all the studies, data were fitted by a two-compartment model. Weight, age, creatinine clearance (CrCL), use of CIS, and number of cycles were commonly added as clearance (CL) covariates [8,14,15,16]. Height, weight, and hepatic enzymes were used to explain variability in the volume of the central compartment [10,14]. In peripheral compartment parameters, body surface area (BSA) was often used as a covariate of inter-compartmental clearance and peripheral compartment volume [7,12,13,14,15,16,17]. Some models also describe the inclusion of genetic covariates since MTX is a substrate for membrane transporters [18]. Moreover, the use of these models in different populations can be challenging, as gene expression may vary across different ethnicity.

Henz and coworkers (2023) built the first population pharmacokinetic model (POPPK) for MTX in Brazilian pediatric patients (median age 6.42 y.o.) with acute lymphoblastic leukemia (ALL) who received doses in the 0.25 to 5 g/m^2^ range. Serum creatinine (SCR), height, blood urea nitrogen (BUN), and a low body mass index stratification (according to the z-score defined by the World Health Organization) were included as clearance (CL) covariates. Genetic variants of genes that encode ATP-binding cassette (ABC) transporters, solute carrier (SLC) transporters, methylenetetrahydrofolate reductase (MTHFR), and glutathione S-transferase (GST) enzymes were investigated as covariates in pharmacokinetic parameters, but the authors did not find any influence of these genetic-related variants on MTX disposition. This model for Brazilian pediatric patients included, for the first time, BUN as a significant covariate for CL, emphasizing the importance of developing models for each population of patients and their specific characteristics [18].

According to the National Childhood Cancer Registry Explorer, the ALL incidence is four times greater in children between 1 and 4 years when OS incidence has a bimodal age distribution, with the first peak during adolescence [19,20]. The patient profiles between these two malignancies differ due to their distinct body composition and liver maturation. Moreover, the dose used to treat OS (12 g/m^2^) is 2.4 times higher than that used in ALL (5 g/m^2^). Patients who received higher doses of MTX tend to present a higher CL, or even a nonlinear CL, as reported previously [18,21].

In this context, in the present work we aimed (1) to investigate if some of the POPPK models published could be used to describe MTX serum concentrations obtained in Brazilian pediatric patients with OS treated according to the BOTG protocol; (2) to develop a POPPK model for MTX in Brazilian pediatric patients with OS treated at Hospital de Clínicas de Porto Alegre (HCPA, Porto Alegre, Brazil), a southern Brazilian reference public hospital, using retrospective TDM data; and (3) to use the built model to predict a high concentration of MTX to anticipate the proper LCV rescue dose to avoid toxicity. To our knowledge, this is the first POPPK model for Brazilian and Latin-American OS pediatric patients.

## 2. Materials and Methods

### 2.1. Study Population

HCPA Ethics in Human Research Committee approved this study (HCPA 4.260.110) and written informed consent was obtained from all participants/legal guardians. This study included 32 Brazilian pediatric patients (mean age 13.25 y.o.), diagnosed with OS admitted by the Pediatric Oncology Service of HCPA between January 2015 and March 2023, who followed the BOTG protocol during cancer treatment. Each patient received 12 cycles of HDMTX by a 4-h infusion at a median dose of 11.9 g/m^2^ (ranging from 5.9 g/m^2^ to 12.9 g/m^2^ depending on the patient’s clinical condition). Patients received proper hydration fluids (3000 mL/m^2^/day) and sodium bicarbonate (45 mEq/m^2^) for urine alkalization before (200 mL/m^2^) and during (500 mL/m^2^) each cycle.

Retrospective patients’ data were included if the cycle had at least 2 observed MTX concentrations and patients were monitored for up to 96 h. Observations beyond 96 h were removed from the data because in extensive TDM patients are selected for dialysis, and this could interfere with CL estimation. Demographic and biochemical variates such as weight, age, height, BSA, body mass index (BMI), gender, serum creatinine (SCr), BUN, aspartate aminotransferase (AST), alanine aminotransferase (ALT), urinary pH, and hematocrit were collected according to each cycle. The data sampling time was lacking in 65.3% of the MTX observations. The same technique described by Henz and coworkers (2023) to input sampling time and guarantee richer data was used in the present study. Briefly, the mean difference between the time of sample arrival at the hospital laboratory and the time of sampling reported for other patients was used to perform a single imputation [19].

MTX monitoring consisted of evaluating its concentrations at pre-determined time points to avoid toxic levels, guiding the administration of the rescue therapy with LCV. According to the BOTG protocol (Table 1), MTX serum concentration should be lower than 10 μM 24 h after infusion, lower than 2 μM 48 h after dosing, and lower than 0.3 μM 72 h after dosing. Serum concentration monitoring should be followed until MTX levels are lower than or equal to 0.3 μM. The standard rescue therapy consists of i.v. bolus administration of 15 mg LCV 24 h after the end of 4 h of MTX infusion, followed by 15 mg orally every 6 h. If moderate or severe toxicity is reported, the proper dose of LCV is increased.

### 2.2. Review and External Validation of Literature Models

A literature search for POPPK models of MTX in patients with OS was performed using PubMed (up to 30 October 2023) with the search terms population pharmacokinetics and methotrexate and osteosarcoma in the title/abstract. Studies were selected for further analysis if they involved POPPK of MTX in patients with OS and if the covariates in the final models matched those in our data. POPPK models of MTX were excluded from external validation if (1) essential information (e.g., typical PK parameters and inter-individual variability) was insufficient for model recompilation; (2) the mean or median age of the patient group was outside the range of 5.0–17.0 years; or (3) there were ethnic differences compared to our patient cohort (e.g., Asian). For qualifying POPPK models, the following details were extracted from the original research: structure of the compartmental model, population PK estimates, covariate model, inter- and intra-individual variability, residual variability, and estimation method.

The studies that fitted the criteria were separately implemented in NONMEN software, with their respective compartments and covariate equations. PK parameters were set to the published values for each model. The predicted concentrations were obtained using the $ESTIMATION routine with MAXEVAL = 0 and POSTHOP options in the software, which means that no evaluations were performed to find the best fit [13]. To evaluate the predictive performance of the model, the prediction error (PE; in percent; Equation (1)), the mean relative error (MPE; in percent; Equation (2)), the median absolute prediction error (MAPE; in percent; Equation (3)), and the relative root mean squared error (RMSE; in percent; Equation (4)) were calculated for population and individual predictions to quantify the predictive performance.
(1)$$\mathrm{PE}\;\left(\%\right)=\frac{C_{i,\mathrm{pred}}-{\;C}_{\mathrm{obs}}}{C_{\mathrm{obs}}}\times 100$$
(2)$$\mathrm{MPE}\;\left(\%\right)=\frac{1}{N}\overset{N}{\underset{i=1}{\sum }}\mathrm{PE}$$
(3)$$\mathrm{MAPE}\;\left(\%\right)=\mathrm{median}\;\mathrm{of}\;\left|\mathrm{PE}\right|$$
(4)$$\mathrm{RMSE}\;\left(\%\right)=\sqrt{\frac{1}{N}\overset{N}{\underset{i=1}{\sum }}{\left(\mathrm{PE}\right)}^{2}}$$
where N is the number of observed MTX concentrations and C_i,pred_ is the individual or population prediction of MTX. Moreover, model-based prediction bias was visually evaluated by plotting the distribution of PE obtained by each POPPK model evaluated. The precision of MTX predictions was considered satisfactory when the percentage of PE within 20% (F20) and 30% (F30) was ≥35% and ≥50%, respectively, and MPE and RMSE were ≤30%. In general, the closer to zero the MAPE and RMSE, the higher the precision of the predictions. If a model reaches these criteria it is considered to be clinically acceptable [13].

### 2.3. Population Pharmacokinetic Analysis

Data were analyzed by a population approach using the nonlinear mixed-effect modeling software program NONMEN (version 7.4, ICON Development Solutions, Ellicott City, MD, USA) and PsN version 4.9.0 software (Perl-speaks-NONMEM, Uppsala, Sweden). The model’s parameters were estimated using the first-order conditional estimation method and interaction (FOCE-I). Ggplot and lattice libraries for R program, version 4.1.2 and RStudio, version 2023.09.1.494 (The R Foundation for Statistical Computing, Vienna, Austria), were used to process data.

Different structural models were examined, including 2 or 3 compartments and either linear or nonlinear elimination methods, to derive the PK parameters. Diverse error models, encompassing proportional, additive, and mixed types, were assessed to characterize the residual variabilities. Exponential assumptions were made for between-subject variabilities (BSVs), while an investigation into between-occasion variability (BOV) was also conducted. Because we used retrospective data in model building, not all the patients’ cycles were available on charts. Despite non-sequential cycles, BOV were coded according to Ho Hui et al. (2019) [17].

MTX concentrations were quantified by chemiluminescent microparticle immunoassay (Alinity i Methotrexate Reagent Kit 09P48) in the hospital laboratory, with a low limit of quantification (LLOQ) of 0.013 mg/L. There were no observations below LLOQ.

The analysis of available covariates consisted of visual inspection followed by a formal evaluation using a stepwise forward additive approach (*p* < 0.05) followed by backward elimination (*p* < 0.01) and biological plausibility. The analysis was performed using the linear, power, and exponential functions normalized by their respective medians. Covariates tested were weight, age, height, body surface area, body mass index, sex, race, serum creatinine, creatinine clearance, hepatic enzymes, hematocrit, and hemoglobin. Period pre- or post-operation was also evaluated as a dichotomous covariate to distinguish cycles before and after tumor resection. Continuous covariates that changed over time were evaluated according to Wählby et al. (2004) [22]. Missing continuous covariates were replaced by the nearest value by date for the patient [23]. The covariates were included one by one in the PK parameters. The covariates included for analysis were those in which the decrease in OFV was greater than 3.84. The order defined for inclusion was based on the lowest OFV value compared to the structural model. Afterward, the inclusion was made sequentially considering the pre-determined order followed by the backward elimination. All patients used the same chemotherapy protocol, so it was not possible to evaluate medical interactions between anticancer drugs.

Model evaluation was performed using a visual predictive check (VPC), the goodness-of-fit (GOF) plots, and analysis of shrinkage (<30%) and %RSE. A thousand profiles were simulated according to the model and arranged in graphs represented by the 5th, 50th, and 95th percentiles together with the experimental data. The robustness of the model was assigned by the bootstrap method (*n* = 1000), which involved comparing the original model estimates with the bootstrap median parameter values and their associated 95% confidence intervals.

### 2.4. Rescue Therapy Guidance

To evaluate the prediction ability of the model, we simulated 1000 profiles for MTX with a standard dose (12 g/m^2^) to predict MTX concentration right after the end of infusion and after 24, 48, and 72 h. The population used for the simulation was created with different levels of renal injury. The normal values of SCr for pediatrics were taken from the proposed guidance of renal impairment for pediatrics [24]. Low, moderate, and high acute kidney injury (AKI) were calculated as described in Equations (5)–(7).
(5)$${\mathrm{AKI}}_{\mathrm{LOW}}=1.5\times \;\mathrm{Reference}\;\mathrm{Scr}$$
(6)$${\mathrm{AKI}}_{\mathrm{MODERATE}}=2.5\times \;\mathrm{Reference}\;\mathrm{Scr}$$
(7)$${\mathrm{AKI}}_{\mathrm{HIGH}}=3\times \;\mathrm{Reference}\;\mathrm{Scr}$$

Then, we overlapped the simulated MTX exposures of low, moderate, and high acute AKI onto the toxicity levels described in BOTG protocol. The threshold concentrations for toxicity are summarized in Table 1.

## 3. Results

### 3.1. Study Population

Considering the period of data collection, 216 cycles of MTX fitted the inclusion criteria. Despite the huge number of cycles, data were quite sparse and only 147 cycles were monitored up to 72 h after the end of the infusion. A total of 563 MTX serum concentrations were included in the dataset used for model building. In the first 24 h after the end of the infusion, only 1% of the observations presented moderate to severe toxic MTX concentrations. All details of patients’ characteristics, sampling, and biochemical parameters are listed in Table 2. Around 78% of the patients were white, 18% black, and 3% were from another ethnicity. Gender was well distributed in this population, comprising 18 males and 14 females. To input missing time samples, the mean difference between the time of sample arrival at the hospital laboratory and the time of sampling reported for patients was 32.5 ± 4.7 min.

### 3.2. Review and External Validation of Literature Models

After literature research, we found one model that could be suitable for our data. The flow of the search process can be seen in Appendix A. The models not selected for evaluation were models that considered genetic covariates [28], were performed in adults [23,29,30] or were conducted in Asian children [17].

The model reported by Colom et al. (2010) [14] described MTX concentrations using a two-compartment model, with IOV in CL and IIV in CL, Vc, and residual error. The significant covariates they used to better fit the data were age and weight. For their analysis, a small number of Spanish patients (*n* = 14), with a mean age of 14.8 years, were used for model development. The comparison of the concentration–time profiles between Spanish and Brazilian patients is shown in Appendix A.

Using the POPPK model reported by Colom et al. (2010) [14], the determination of PE evidenced an important bias in the population predictions of Brazilian patients (Figure 1). The results of overall bias and precision for the model evaluated were a MPE of 81.42%, a MAPE of 76.96% (F20 12.1% and F30 18.6%), and an RMSE of 245.03%. All the results extrapolated the acceptance criteria, indicating that the model was not adequate to describe Brazilian pediatric patient’s pharmacokinetics.

### 3.3. Population Pharmacokinetic Analysis

MTX concentrations versus time data used to build the POPPK model are shown in Figure 2. A two-compartment model with first-order elimination from the central compartment parametrized in terms of clearance (CL), central compartment volume (Vc), peripherical compartment volume (Vp), and inter-compartmental clearance (Q) best described the data. Models incorporating nonlinear elimination and three compartments disposition were investigated, but the OFV did not vary significantly (OFV values of −1513.43 and −1515.56, respectively) compared with the linear two-compartment model (OFV value of −1513.84), which led to imprecision in the estimated parameters.

BSV was exponentially added in CL (19.8%) and Vc (13.5%), with a correlation of 94%. The inclusion of exponential BOV in CL (15.1%) decreased the OFV by 596.97 points, indicating the significance of this parameter for the model. Between-occasion variability was estimated as a single value for all different MTX occasions. The residual variability was described by a proportional error model.

The covariate analysis showed that the inclusion of serum creatinine (SCr) as a covariate on CL leads to a significant decrease in the OFV (ΔOFV: 23.27, *p* < 0.001) and helped to explain both variabilities associated with this parameter, decreasing BSV and BOV by 2.5% and 3.7%, respectively. The use of CrCL as a covariate on CL led to a similar decrease in the OFV as SCr, indicating that the primary biomarker was sufficient to explain the variability. The investigation of height as a covariate for CL, which is also used in the Schwartz equation to calculate CrCL, had no impact on BSV or BOV. To explain Vc variability, BSA was added, leading to a variability drop of 11.3% (ΔOFV: 6.75, *p* < 0.01) (Appendix A). The final model individual parameter equations including covariates are expressed as follows:(8)$$\mathrm{CLi}=14.8\times {\left(\frac{\mathrm{SCr}}{0.58}\right)}^{-0.192}\times {\;e}^{\left(0.0391+0.0228\right)}$$
(9)$$V_{C}i=82.5\times {\left(\frac{\mathrm{BSA}}{1.45}\right)}^{0.301}\times {\;e}^{\left(0.0181\right)}$$
(10)Qi=0.178
(11)$$V_{p}i=5.72$$

All parameters for the final POPPK model are shown in Table 3, with the non-parametric bootstrap results. The PK parameters are all estimated with good precision, which can also be seen through the goodness-of-fit plots in Figure 3. The diagnostic plot showed good agreement between predicted and observed concentrations and no tendencies in residual plots. The VPC plot can be seen in Figure 4, and it confirms the predictive performance of the model.

### 3.4. Rescue Therapy Guidance

Managing HDMTX therapy in patients with OS requires careful monitoring and intervention to minimize toxicity while ensuring therapeutic efficacy. We performed simulations in a virtual population with ages ranging from 5 to 18 years old and three different stages of AKI (low, moderate, and high). The model was able to predict toxic MTX concentrations over 72 h after the end of infusion in different stages of renal impairment in children older than 5 years. The ability to predict toxic levels of MTX before starting the infusion of high doses (12 g/m^2^) allows adjusting appropriate doses of LCV to avoid adverse effects, once the LCV dose starts before the first measurement of MTX.

## 4. Discussion

The present study describes the first POPPK model for MTX in Brazilian pediatric patients with diagnosis of OS to guide rescue therapy in a routine clinical setting, while highlighting the effects of BSA and AKI on MTX exposure. To the best of our knowledge, based on population similarity, disease, and available covariates, the POPPK model described by Colom et al. (2009) [14] had the potential to fit our data. However, the external validation of the previously reported model led to unprecise predictions of our population MTX pharmacokinetics.

A previous study from our group described the result of improper hydration, measured by the ratio between SCr and BUN, on MTX concentrations in pediatric patients with acute lymphoblastic leukemia (ALL) [18]. However, the previously reported model did not describe the present data, possibly because the patients’ median age (6.42 y.o.) and range (0.33 to 17.8 y.o.) was different from the median (13.25 y.o.) and range (5 to 18 y.o) of the age of the patients of the present study. Furthermore, the dose used for treating ALL (5 g/m^2^) is smaller than the dose (12 g/m^2^) used in OS treatment.

From the 32 patients included in the present study, 24 were adolescents (>12 years). MTX therapy in OS adolescent patients presents a low risk of toxicity [31]. This was confirmed by our analysis when adolescent patients did not have a significant number of toxic events reported (1%). Furthermore, patient’s age was not well distributed around all childhood lifetime stages, which probably explains why, in the current POPPK model, patient’s age was not included as a covariate to explain PK parameter variability, as described in previous studies [8,14].

The CL estimated in the present study (15.1 L/h) and the distribution parameters Vc (82.5 L), Vp (5.72 L), and Q (0.178 L/h) agree with those reported by Nader et al. (2017) [32], who studied adolescents and young adults with hematological malignancies. Pharmacokinetic parameters for MTX reported in children and young adults with OS are lower than those estimated in the present study. The estimated parameters reported previously were a CL ranging between 3.68 and 5.81 L/h, a Vc ranging between 11 and 19 L, a Q between 0.01 and 0.03 L/h, and a Vp between 0.4 and 0.5 L [14,15,16]. The protocol of those studies had a sampling right after the end of the infusion, which means that they caught the MTX serum peak of concentrations. On the other hand, in the BOTG protocol followed in the present study, sampling time started 24 h after the end of the infusion, making it impossible to capture the MTX serum peak, leading to an estimation of higher Vc. Furthermore, given that patients who receive HDMTX often require additional hydration and urine alkalization, these individuals may demonstrate increased clearance [17]. The CL estimated in the present study was 2-fold higher than that estimated by Henz et al. (2023) [18] for Brazilian ALL pediatric patients who received ~5 m/kg doses. This is probably because the ALL study demonstrated that patients were frequently dehydrated, which was not observed in the present study, and this could lead to a slower CL.

To explain CL variability, SCr was added as a covariate in a power function. Renal function was associated with MTX CL in OS [10,29] as well as in ALL [17,33] patients. This is coherent since the MTX elimination pathway is mostly renal [34,35,36]. BSA was an important element in the present analysis, being selected as a significant covariate for Vc, similarly to the approach described in previous models [17]. As reported before, BSA presents a direct relationship with the volume of distribution, especially when high amounts of drug are administered [18]. 

Hepatic enzymes [10], urine pH [37], and hydration [38] were important covariates used to explain MTX PK in other populations. In the population used to build the present model, liver enzymes varied largely (from 6–8 to 1052 U/L), preventing the determination of their impact on PK parameters. On the other hand, urine pH values were mostly alkaline over the treatment period in all patients, impeding their use to explain CL variability among patients or cycles. Full-body hydration helps MTX depuration, avoiding the maintenance of high serum concentrations. Differently from our model for MTX in ALL patients, in the present model, BUN did not explain MTX variability in CL, probably because 95% of the patients were well hydrated throughout the treatment.

In the present study, the influence of genetic polymorphism on MTX distribution transporters or elimination enzymes was not investigated. Although two models previously reported have included genetic polymorphism on ABCG2 and SLCO1B1 as covariates on CL, both studies conclude that the major covariate responsible for explaining the variability in this parameter was body weight [28,39]. Additionally, the two studies conducted with Latin American pediatric patients that evaluated different genetic variants related to MTX disposition found no significant influence of genes on MTX pharmacokinetics [18,40]. Finally, the inclusion of genetic polymorphism as a covariate in POPPK models used to adjust doses of highly variable drugs, such as MTX, in low–middle-income countries like Brazil is only justifiable if the covariate accounts for a significant percentage of the parameter variability and its determination is part of the clinical routine of the service where the model will be used.

Estimating BOV is important in long-duration therapies, especially in chemotherapy, as already described in previously reported POPPK models [14,17,28]. However, the determination of BOV using sparse data is challenging. These studies justify why some published MTX models were not able to estimate BOV due to a lack of observations in different cycles, especially when using retrospective data [35,36]. In the present study, we were able to estimate the BOV variability of MTX cycles individually due to the large number of cycles (*n* = 21) included in the final dataset.

In patients with OS, the achievement of C_max_ values around 700–1000 μM is associated with prolonged disease-free survival, tumor necrosis, and longer improvement of MTX efficacy [9,35]. On the other hand, the drug peak serum concentration was not found to be a factor for delayed MTX CL or toxicity levels in pediatric patients [36]. Therefore, it is important to reach a certain level of MTX in plasma to maximize effectiveness. Thinking on that, we focused this work on adjusting the LCV amount but not reducing the MTX dose. Thus, we did not compromise MTX’s C_max_ and improved treatment performance.

Although the BOTG protocol was followed with the patients with OS in the present study, sometimes patients present moderate to severe toxicity, needing an LCV dose adjustment after the first 24 h. Using the model individual parameters, we can estimate MTX serum concentrations before chemotherapy administration. The POPPK model developed was used to simulate MTX concentration in different stages of AKI. We demonstrate that patients above 10 years with any degree of AKI present a higher probability of achieving severe toxic MTX concentration (Figure 5). On the other hand, patients from 5 to 10 years of age present higher risk of moderate toxic concentrations of MTX in high and moderate grades of AKI. Since the first dose of LCV is administered before the first MTX concentration measurement, with this model, we can predict whether the patient will have a toxic response and thereby anticipate the correct dose of LCV even before the first measurement. Therefore, we believe that, by using this model, we could prevent adverse effects.

The limitations of the present study include the use of sparse data, due to its retrospective nature and reliance on routinely collected hospital data. Additionally, there was a lack of sampling time records for MTX dosing in some patients and no time points in the peak of concentration. Furthermore, a low number of patients were included in the study. All of this precluded the possibility of conducting external validation of the model. In this scenario, the present study paves the way for a prospective study to explore the practical application of the developed POPPK model in a clinical setting, allowing the incorporation of additional patients with OS, and thus potentially refining the model’s predictive accuracy.

The routine application of MIPD in Brazil is very limited or practically nonexistent. The translation of the POPPK model into clinical PK equations facilitates its integration into clinical practice, enabling the prediction of toxic concentrations in any pediatric patient diagnosed with OS. We worked with patients admitted to a reference pediatric cancer center that receives patients from all the southern regions of Brazil, which makes the POPPK model developed potentially suitable for describing and predicting MTX concentrations in pediatric patients in the south of the country. However, the application of this model for cross-center use should be evaluated since Brazil is a country with significant ethnic diversity, which results in regional variation in patient’s profiles.

In hindsight, this study provides evidence for the adjustment of LCV doses in HDMTX therapy for patients with OS. Furthermore, it underscores the notion that empirically guided dosing may have adverse effects on the treatment of pediatric patients. Thus, healthcare professionals can ensure a more effective and safer anti-cancer therapy by combining the model predictions and TDM for southern Brazilian patients.

## 5. Conclusions

We described a POPPK model for MTX built on retrospective data from TDM of southern Brazilian children with OS. The model showed a significant contribution of SCr and BSA in explaining the variability observed in MTX CL and Vp, respectively. The clinical usefulness of this model in managing HDMTX pharmacotherapy in children is in the prevention of adverse effects by anticipating LCV rescue doses. The routine application of the POPPK model could improve the management of MTX therapy in OS pediatric patients.

## Figures and Tables

**Figure 1 pharmaceutics-16-01180-f001:**
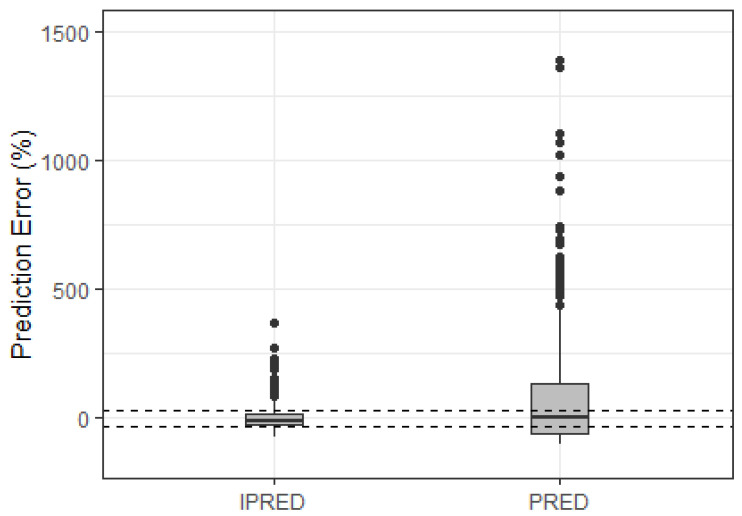
Prediction error (PE) distributions of the MTX POPPK model evaluated in the external dataset. Black solid line, PE equal 0 (unbiased); dashed black lines, PE equal to ±30% (acceptable bias).

**Figure 2 pharmaceutics-16-01180-f002:**
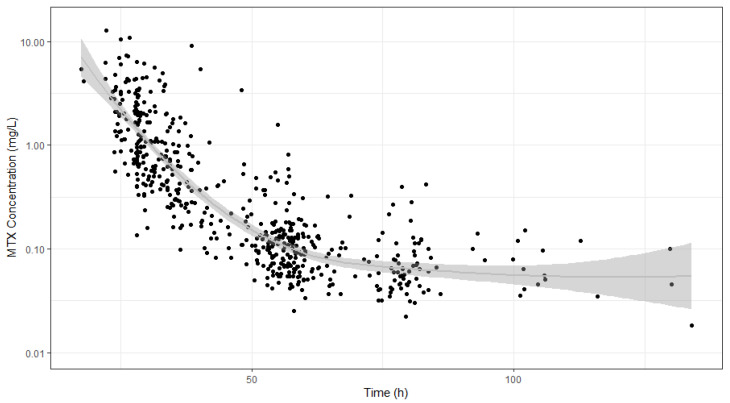
Serum MTX concentrations (mg/L) vs. time (h) after the end of 4 h of infusion (h). Black dots are the measured concentration points, the gray line is a Loess line indicating the trend of the data, and the gray area is the 95% confidence interval.

**Figure 3 pharmaceutics-16-01180-f003:**
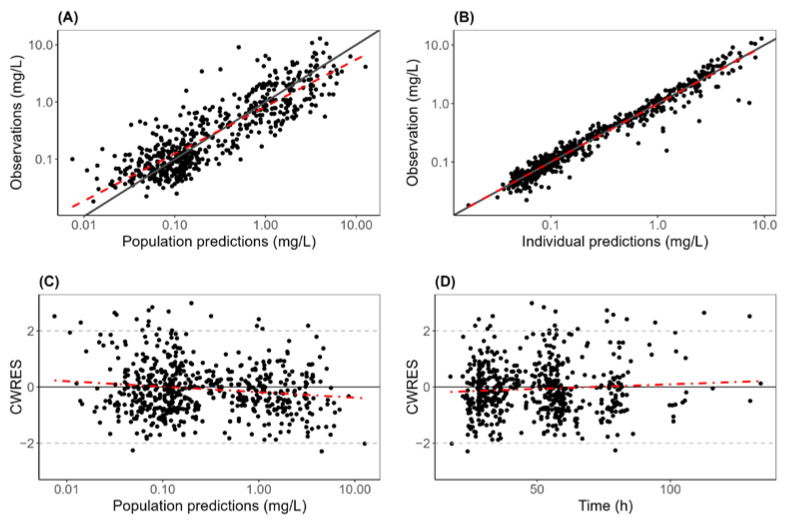
Goodness-of-fit plots of the final MTX POPPK model. (**A**) Correlation between observations and population predictions by the final model. (**B**) Correlation between observations and individual predictions by the final model. (**C**) CWERS (conditional weighted residuals) vs. individual prediction distributed around slope zero. (**D**) CWERS vs. time, showing no major bias in the model.

**Figure 4 pharmaceutics-16-01180-f004:**
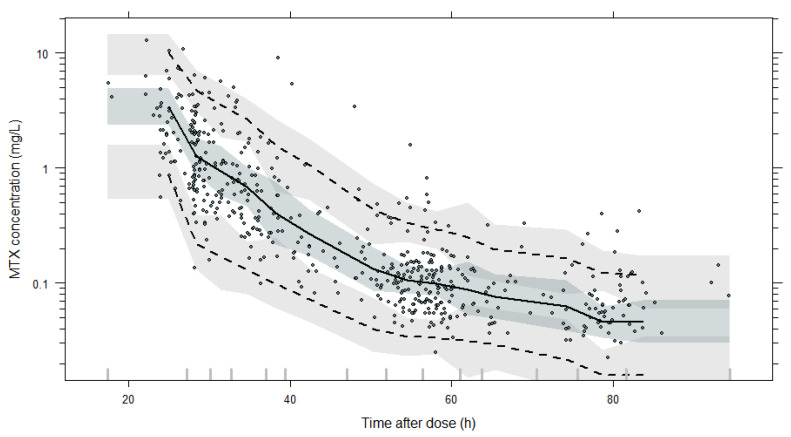
Visual predictive check of the final POPPK model. VPC is based on 1000 simulations and shows a comparison of the observations (dots) with the 10th, 50th, and 90th percentiles of simulated profiles (shadow areas) and of observations (dashed and solid lines).

**Figure 5 pharmaceutics-16-01180-f005:**
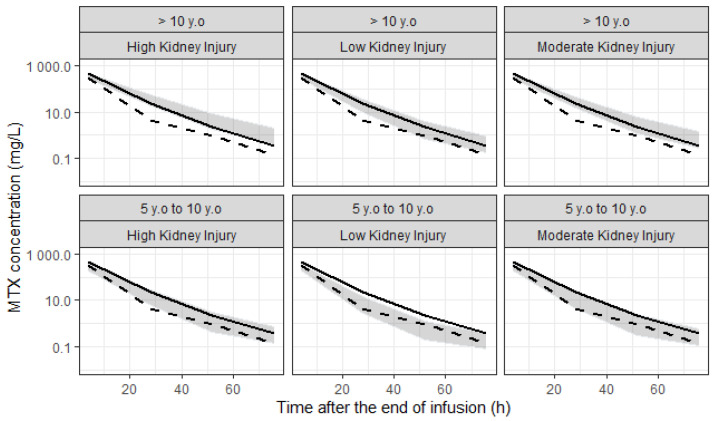
Simulated profiles for MTX concentration right at the end of infusion and 24, 48, and 72 h after. Profiles are expressed as their 95%CI (gray area), in two different age groups for three levels of acute kidney injury. Severe and moderate toxicity limits are represented by the continuous and dashed lines, respectively.

**Table 1 pharmaceutics-16-01180-t001:** Serum concentration thresholds to MTX toxicity and LCV dose recommendation according to the BOTG protocol.

MTX Level		Target MTX Levels to Avoid Toxicity		Moderate MTX Toxicity	Severe MTX Toxicity
C_24 h_		<10 μM (4.5 mg/L)		10–50 μM (4.5–22.7 mg/L)	>50 μM (22.7 mg/L)
C_48 h_		<2 μM (0.9 mg/L)		2–5 μM (0.9–2.3 mg/L)	>5 μM (2.3 mg/L)
C_72 h_		<0.3 μM (0.1 mg/L)		0.3–1 μM (0.1–0.5 mg/L)	>1 μM (0.5 mg/L)
**LCV**	first 24 h	15 mg i.v. bolus	30 mg i.v. bolus		150 mg i.v. q3 h (until 1 μM)
	after 24 h	15 mg v.o. q6 h	30 mg v.o. q6 h		15 mg i.v. q3 h

**Table 2 pharmaceutics-16-01180-t002:** Patients’ demographic characteristics.

Demographic Data	Unit	Value
Number of patients		32
Race		
White	%	78.1
Black	%	18.8
Other	%	3.1
Sex	M/F	18/14
Age (years)	Median (Range)	13.25 (5–18)
Weight (kg)	Median (Range)	47 (13.80–85.50)
Height (cm)	Median (Range)	159 (115–177)
Body surface area ^a^ (m^2^)	Median (Range)	1.45 (0.67–2.03)
Body mass index ^b^ (kg/m^2^)	Median (Range)	18.11 (10.43–28.56)
Treatment		
Dose (g/m^2^)	Median (Range)	11.9 (5.9–12.9)
Total of cycles		216
Cycles pre-tumor ressection		96
Cycles post-tumor ressection		120
Total number of MTX concentrations		563
MTX concentrations/cycle	Median (Range)	3 (2–6)
Clinical data		
SCr (mg/dL)	Median (Range)	0.58 (0.17–3.2)
CrCL ^c^ (mL/min/1.73 m^2^)	Median (Range)	190.28 (37.41–372.05)
BUN (mg/dL)	Median (Range)	17 (4–87)
AST (U/L)	Median (Range)	34 (8–1052)
ALT (U/L)	Median (Range)	47 (6–1052)
Urinary pH value	Median (Range)	7.5 (7–9)
Hematocrit (%)	Median (Range)	28.9 (10–52.20)
Hemoglobin (g/dL)	Median (Range)	9.6 (3.28–13.2)

SCr: serum creatinine; CrCL: creatinine clearance; BUN: blood urea nitrogen; AST: aspartate aminotransferase; ALT: alanine aminotransferase; ^a^: calculated with Haycock equation [25]; ^b^: calculated with Quetelet equation [26]; ^c^: calculated with Schwartz equation [27].

**Table 3 pharmaceutics-16-01180-t003:** Population pharmacokinetic parameter estimates for the MTX POPPK model.

Parameter		Unit	Estimate	R.S.E (%)	Bootstrap Medians * (95% CI)
Clearance (CL)	TVCL	L/h	14.8	19	14.4 (10.7–19.3)
	ω_BSV_	%	19.8	16	20.2 (14.1–27.5)
	ω_BOV_	%	15.1	6	14.8 (12.8–16.9)
	θ_SCr_		−0.192	31	−0.201 (−0.32–−0.08)
Central Volume (Vc)	TVVC	L	82.5	23	79.9 (54.5–115.9)
	ω_BSV_	%	13.5	29	15.3 (7.6–25.1)
	θ_BSA_		0.301	36	0.298 (0.025–0.481)
*Correlation CL–Vc*		%	94		
Peripheral Volume (Vp)	TVV_P_	L	5.72	35	5.42 (3.1–10.5)
Inter-compartmental Clearance (Q)	TVQ	L/h	0.178	31	0.171 (0.106–0.285)
Residual Variability					
Proporcional Error		%	30.9	4	30.3 (27.5–33.1)

* 853/1000 successful runs.

## Data Availability

The raw data used in this study are available upon direct request to the authors.

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
