# Peer review of "Anticipating Leucovorin Rescue Therapy in Patients with Osteosarcoma through Methotrexate Population Pharmacokinetic Model"

_pharmaceutics, 2024, doi:10.3390/pharmaceutics16091180_

Round 1

Reviewer 1 Report

Comments and Suggestions for Authors

The objectives of the study by Ben Olivio, et al. were (i) to externally validate previously published population PK models of MTX in pediatric patients with OS, (ii) to develop a population PK model of MTX in pediatric patients with OS treated with the BOTG protocol, and (iii) to evaluate whether the developed model can be used to predict LCV rescue dose to prevent toxicity. The objectives of the study seem reasonable, and indeed the application of the developed MTX population PK model in pediatric patients with ALL is problematic because the two patient populations differ in age. In addition, the treatment protocols differ and higher MTX doses are used in patients with OS. 

Specific comments:

1. Introduction: at the end of the introduction section it states “To our knowledge, this is the first POPPK model for Brazilian and Latin American OS pediatric OS pediatric patients”. Explain more clearly why it is important to develop an HDMTX-POPPK model for this specific population. You provide some hints, e.g., different treatment protocols, higher MTX dose, genetic factors (SLC1B1 and ABCG2); however, I think this should be more clearly articulated for the average reader. It would also help the reader understand why the PK studies in the Asian population were excluded from external validation.

2. The concept of first person should be adhered to throughout the manuscript. For example, OS patients need to be rephrased as patients with OS.

3. Methods: you report that 65.3% of MTX measurements were missing sampling times. This is quite a lot and seems critical, especially for a drug with relatively rapid distribution and elimination kinetics like MTX. The missing sampling times were imputed with the calculated times based on the mean difference between the sampling time and arrival at the laboratory for the remaining 45% of samples for which sampling times were reported. Please indicate the mean difference and its range of variation. This will give the reader an idea of the magnitude of the possible error introduced by the imputation of sampling times. Have you performed a sensitivity analysis for the imputation method?

4. Methods: you discarded MTX cycles with prolonged TDM beyond 96 hours. In my opinion, you introduced a bias in this way as only cycles with slow MTX elimination were excluded. In my opinion, you should have kept all the data and excluded only the post-dialysis measurements, or perhaps even better, kept all the data and modeled MTX clearance during dialysis. It is also not clear why cycles with 1 concentration measurement were excluded. These measurements could have been used for model development or external validation of the a priori (population) prediction.

5. Methods (2.2.): Please explain why studies in the Asian population were excluded from external validation!

6. Methods (2.3.): What type of nonlinear models were evaluated?

7. Lines 206-209: you cannot estimate model accuracy by bootstrap! It can only be used to estimate the precision of the parameters and the robustness of the model.

8. Section 2.4: You should describe more clearly how the simulation study was performed. You could have used your final model and run simulations in NONMEM ($SIM ONLYSIM nonmem option) for different populations (patients with elevated Scr 1.5, 2.5 and 3 times the reference range). What type of distribution did you use for the Scr reference range? I assume the Scr reference range is also age dependent?!

You also state that the phase inversion of the distribution elimination is at 0.5 mM. How did you validate this assumption and why was this assumption necessary? You do not need to make this assumption for the simulation. Equation S1 is only valid when the infusion reaches the steady state. I doubt that the steady state will be reached within 4 hours after the MTX infusion.

9. Results (3.2.): Show more data on the inadequacy of the literature model! Instead of Figure S2, I would welcome a VPC plot for the literature model. (I am also interested in the difference in the variability of MTX PK)

10. Results (3.3.), line 276: It is impossible to get a higher OFV with the 3-compartment model and a nonlinear 2-compartment model than with a linear 2-compartment model. Both models could converge to the linear 2-compartment model, so that a lower or at least the same OFV can be expected. It could be that the two models do not converge (insufficient data for more complex models) or that they were trapped in a local minimum.

11. Estimates of the model parameters (Table 3): It would be useful to standardize the parameters to a typical adult weight of 70 kg. In this way, it is easier to compare the parameters of different models (ALL vs. OS) and different populations.

12. Was BSA a significantly better covariate than patient weight?

13. The estimated covariate effect of BSA on peripheral compartment volume makes no sense (is not biologically plausible)! It predicts a decrease in distribution volume with an increase in BSA! I suspect the reason for this is the correlation between the parameters (CL, Vc, VP and Q) as they all correlate with body size. Have you tried estimating the off-diagonal elements of the omega matrix (BSV and BOV)?

14. Results (3.3.): The results for predicting LCV recovery are poorly reported. How accurate was the a priori (population) and a posteriori (individual) prediction of LCV dose at different time points? How were the individually predicted MTX concentrations calculated? Did you use the "Proseval" procedure in PSN?

15. Figure 5: The results are obvious and well known, i.e. increase in MTX concentration with decrease in renal function. The figure is difficult to recognise.

16. Discussion: The discussion section is lengthy and should be structured more sensibly.

17. There are many typos and the language should be improved (line 6, line 24: Data were instead of Data was, …), just to name a few.

Comments on the Quality of English Language

There are many typos and the language should be improved (line 6, line 24: Data were instead of Data was, …), just to name a few.

Author Response

Dear Reviewer, find attached the answers to you comments,  Best wishes, Bibiana

Reviewer 2 Report

Comments and Suggestions for Authors

The manuscript summarizes population pharmacokinetic (PopPK) analysis of methotrexate (MTX) serum concentrations in Brazilian pediatric patients with osteosarcoma (OS), PopPK model-based simulations of effects of acute kidney injury on the MTX concentrations, and its implications on the leucovorin rescue therapy in pediatric patients with OS. The manuscript is devoted to an important clinical topic, and is overall of good quality. However, I have several points of critique and comments related to the contents of this manuscript.

Specific comments

1. The analytical assay of MTX in serum samples - line 184 -– should be presented in a separate section, with higher level of details on the sample processing, type of assay, apparatus, range, linearity, selectivity, etc.

It is not clear whether the applied assay was able to quantify selectively and reliably the serum concentrations of MTX (vs. metabolites, e.g., 7-OH MTC, DAMPA, …). The choice of the samples and the analyte should be substantiated. Generally, the concentrations of the intracellular polyglutamate MTX metabolites in the T-lymphocytes and/or erythrocytes are expected to be a better marker of MTX treatment safety and efficacy. Please elaborate on this comment in the revised manuscript.

2. Fig. 5 – The data on this figure and their meaning are not clear. Why there was a need to use an arbitrary “phase inversion” concentration of 0.5 uM (equations 8a-10)? Why the simulation was performed for the selected data points only (end of infusion, 24, 48 ,72 hr) and not for all the time course of drug concentrations (0-72 hr, based on the PopPK model and parameters in Table 3)? Why the values on the Y-axis reach 1000 mg/L while the observed data are below 10 mg/L (Fig. 2)? Please check, correct if necessary, and explain. The format/scale of Fig. 5 should be changed to make visible the differences in the lines and shaded area.

Additional comments

Fig. 2, 4, 5 – please check the units – mg/L or mg/mL; please use consistent units in the manuscript (the figures, tables, text).

Fig. 2 – please check the Y axis – is this a logarithmic axis? Please add the minor scale bars.

Line 6 – “1 and 2” – please correct (1,2)

Line 24 – data were described

Line 32 – please correct the grammar

Table 3 – the last line – please correct the word “proportional”

Comments on the Quality of English Language

none

Author Response

(The authors gave the same response as above.)

Reviewer 3 Report

Comments and Suggestions for Authors

Ben Olivo et al. developed a popPK model for high-dose MTX in patients with osteosarcoma. The disease is rare, and the development of such a model adds value to the current knowledge.
In their work,  the authors showed that using a previously reported model built on data from another population may lead to a significant bias in predicted MTX concentrations. Hence, the merit is sound.
The model building is adequate and presented diagnostic graphs show good fit quality without significant misspecifications. It is surprising to see that Vc did not have IIV or IOV elements, but Vp had. Please describe the model-building steps in the Supplementary File,  starting from the base model and using appropriate OFVs.
The study aimed to predict MTX concentrations to guide subsequent rescue LCV therapy. It was not adequately described.
The approach proposed by the Authors was to use hybrid constants and MTX  concentrations at the end of the infusion and 24, 48, and 72 h after completing it. Then, they presented these results in Figure 5. However, these profiles show how kidney injury could impact the MTX  concentrations, but this prediction was not validated; therefore, the sentence in line 320, "The model was able to predict...", is too far-fetched.
What the authors could do is perform some additional calculations - it would be valuable to see the differences in the observed and predicted (using equations 8a - 10)  MTX concentrations in the study population. Metrix such as MAE, MAPE,  RMSE, etc. Another approach would be to use Bayesian methods - fix the final model estimates and supply, e.g., observed Cmax ; then predict values for 24, 48, and 72 h and compare them with the observations. In my opinion, it would be interesting to see how many MTX concentrations are necessary to accurately estimate the risk of MTX toxicity and LCV  therapy.
Lastly, the leucovorin rescue therapy guidance through popPK modeling is a suggestion. It is very promising, but the superiority of such an approach over "classic" TDM has not yet been proved.

Author Response

(The authors gave the same response as above.)

Round 2

Reviewer 1 Report

Comments and Suggestions for Authors

The authors adequately addressed the majority of my concerns. Specifically, of my major concern were Comments 8 and 13.  The authors applied different simulation method and considerably changed their model, as it was suggested.

Still I do not see the reason why they did not try to predict leucovorin dose based on population (a priori) model predictions and individual (a posteriori) model predictions based on the measurements collected before the leucovorine dosing (Comment 14), as suggested by the manuscript title (Guiding leucovorin rescue therapy ...). This is not critical, but in my opinion it would significantly enhance the usefulness of the model for the clinicians.

Comments on the Quality of English Language

There are some minor errors. I recommend text editing and proofreading by a native speaker. 

Author Response

For research article

Response to Reviewer 1

1. Summary

2. Questions for General Evaluation

Reviewer’s Evaluation

Does the introduction provide sufficient background and include all relevant references?

Can be improved

Are all the cited references relevant to the research?

Can be improved

Is the research design appropriate?

Can be improved

Are the methods adequately described?

Can be improved

Are the results clearly presented?

Can be improved

Are the conclusions supported by the results?

Can be improved

3. Point-by-point response to Comments and Suggestions for Authors

Comments 1: The authors adequately addressed the majority of my concerns. Specifically, of my major concern were Comments 8 and 13.  The authors applied different simulation method and considerably changed their model, as it was suggested.

Still I do not see the reason why they did not try to predict leucovorin dose based on population (a priori) model predictions and individual (a posteriori) model predictions based on the measurements collected before the leucovorine dosing (Comment 14), as suggested by the manuscript title (Guiding leucovorin rescue therapy ...). This is not critical, but in my opinion it would significantly enhance the usefulness of the model for the clinicians.

Response 1: Thank you for your inquiry. Our initial approach was to calculate the appropriate dose of leucovorin. Our objective was to assess the impact of administered leucovorin on MTX concentrations, based on the doses given to each patient. However, we did not find detailed and consistent records of leucovorin doses administered to hospitalized patients between 2015 and 2021. Furthermore, the Hospital do not quantify patient´s leucovorin concentrations and we understand that we cannot calculate a proper dose of leucovorin without its pharmacokinetics in this patient´s group. For these reasons we opted to use the model to estimate MTX concentrations in a toxic range and then apply leucovorin dosing regimen proposed in the BOTG protocol, based on estimated MTX concetnrations. We understand your concern that the title of this work could be misleading. Therefore, we decided to change the title of the manuscript to “Anticipating leucovorin rescue therapy in patients with osteosarcoma through methotrexate population pharmacokinetic model” to make it more befitting.

4. Response to Comments on the Quality of English Language: There are some minor errors. I recommend text editing and proofreading by a native speaker. The manuscript´s English was revised.

Reviewer 3 Report

Comments and Suggestions for Authors

The authors have addressed most of my suggestions. The model improved after reparametrization.

Please provide more details on ref. [26], preferably a link to the report file, as it is not easy to find the specific Scr value on the thinkkidneys.uk web page.
I assume the Rescue Therapy simulations accounted for between-subject variability of SCr and BSA -  both significant model covariates. Were BSA and SCr variabilities the  same as in the study? What about the variabilities of low, moderate, and  high AKI? Have the authors assumed that they are the same as for normal  SCr?
It also needs to be clarified why the authors chose the 10 y.o. cutoff for MTX concentration predictions. LCV dosing does not include age as the guiding parameter, only SCr. The median age in the study was 13.25, so the model could not be adequate for the 5 - 10 y.o. range, due to a limited number of subjects.

The sentence in lines 373 - 375 needs to be clarified. What did the authors mean? It is logical that the larger the distribution volume is, the lower the concentrations are. Did the authors mean the non-proportionality?

Lastly, if this model would be used for guided LCV rescue therapy, then - given certain SCr and BSA - should estimate/predict MTX concentrations. Then, perhaps, BSA and not age should be used to estimate the  toxicity risk.

Author Response

For research article

Response to Reviewer 3

1. Summary

2. Questions for General Evaluation

Reviewer’s Evaluation

Does the introduction provide sufficient background and include all relevant references?

Yes

Are all the cited references relevant to the research?

Yes

Is the research design appropriate?

Yes

Are the methods adequately described?

Yes

Are the results clearly presented?

Yes

Are the conclusions supported by the results?

Can be improved

3. Point-by-point response to Comments and Suggestions for Authors

Comments 1: Please provide more details on ref. [26], preferably a link to the report file, as it is not easy to find the specific Scr value on the thinkkidneys.uk web page.

Response 1: Link added in line 543 [https://www.thinkkidneys.nhs.uk/aki/wp-content/uploads/sites/2/2016/05/Guidance-for-paediatric-patients-FINAL-1017.pdf]

Comments 2: I assume the Rescue Therapy simulations accounted for between-subject variability of SCr and BSA -  both significant model covariates. Were BSA and SCr variabilities the  same as in the study? What about the variabilities of low, moderate, and  high AKI? Have the authors assumed that they are the same as for normal  SCr?

Response 2: The SCr reference levels for children are age dependent. We perform normal distribution in R using trucnorm function, with upper and lower boundaries depending on SCr range level. We assumed the same variability from normal SCr.

Comments 3: It also needs to be clarified why the authors chose the 10 y.o. cutoff for MTX concentration predictions. LCV dosing does not include age as the guiding parameter, only SCr. The median age in the study was 13.25, so the model could not be adequate for the 5 - 10 y.o. range, due to a limited number of subjects.

Response 3: We used these 2 different groups because de SCR levels and BSA levels in pediatrics beyond 10 years starts to elevate until reach the adult range.

Comments 4:  The sentence in lines 373 - 375 needs to be clarified. What did the authors mean? It is logical that the larger the distribution volume is, the lower the concentrations are. Did the authors mean the non-proportionality?

Response 4: Thank you for your observation. The word MTX in line 373 was wrong. Tne corrected word was BSA , which presents a direct relationship with Vc, when increases BSA increases Vc.

Comments 5:  Lastly, if this model would be used for guided LCV rescue therapy, then - given certain SCr and BSA - should estimate/predict MTX concentrations. Then, perhaps, BSA and not age should be used to estimate the toxicity risk.

Response 5: Thank you for your question. The reason we separate the patients in age groups was due to the SCr reference levels. But we also perform a normal distribution on BSA values for each range.

Round 3

Reviewer 3 Report

Comments and Suggestions for Authors

The authors have sufficiently answered my comments. I have no further suggestions.